# The Appearance of Osteomyelitis of the Foot and Disseminated Subcutaneous Abscesses During Treatment for Disseminated Tuberculosis Infection in an Immunocompetent Patient: Case Presentation of a Paradoxical Reaction and Literature Review

**DOI:** 10.3390/idr17030046

**Published:** 2025-05-02

**Authors:** Luca Santilli, Benedetta Canovari, Maria Balducci, Francesco Ginevri, Monia Maracci, Antonio Polenta, Norma Anzalone, Lucia Franca, Beatrice Mariotti, Lucia Sterza, Francesco Barchiesi

**Affiliations:** 1Unit of Infection Diseases, San Salvatore Hospital, Azienda Sanitaria Territoriale Pesaro e Urbino, CAP 61121 Pesaro, Italy; benedetta.canovari@sanita.marche.it (B.C.); maria.balducci@sanita.marche.it (M.B.); francesco.ginevri@sanita.marche.it (F.G.); monia.maracci@sanita.marche.it (M.M.); antonio.polenta@sanita.marche.it (A.P.); s1110352@pm.univpm.it (N.A.); s1102402@pm.univpm.it (L.F.); beatrice.mariotti@sanita.marche.it (B.M.); lucia.sterza@sanita.marche.it (L.S.); francesco.barchiesi@sanita.marche.it (F.B.); 2Department of Biomedical Sciences and Public Health, Università Politecnica delle Marche, CAP 60126 Ancona, Italy

**Keywords:** tuberculosis, *Mycobacterium tuberculosis*, foot osteomyelitis, subcutaneous and skin abscesses, paradoxical reaction

## Abstract

**Background:** The appearance of new clinical manifestations (for example, subcutaneous or skin abscesses) during anti-tuberculosis treatment is generally indicative of therapeutic failure. The cause of therapeutic failure may be the presence of a drug-resistant *Mycobacterium* infection or to the failure to achieve a sufficient concentration of the drugs in the bloodstream. **Case report:** Here, we report the case of a 25-year-old man suffering from tuberculosis infection with lymph-node and pulmonary involvement and an atypical response to specific therapy. Two weeks after starting four-drug antitubercular treatment, the patient began to experience fever, pain and functional impotence in the left foot and ankle, with subsequent evidence of ankle and tarsal osteomyelitis. Four weeks after starting treatment, the patient presented with several widespread, painful subcutaneous abscesses on the trunk, back and right lower limb. Drainage was performed from the ankle and from one of the abscesses, and polymerase chain reaction (PCR) showed a positive result for *M. tuberculosis* in both samples, with the absence of resistance to drugs. Anti-tubercular medications were continued, with resolution of the pulmonary and bone involvement but with persistence of subcutaneous abscesses, although subsequent drainages showed the absence of mycobacterium tuberculosis. **Conclusions:** We describe an unusual presentation of paradoxical reaction in the form of osteomyelitis and subcutaneous abscesses in an immunocompetent TB patient, and we reported other similar cases of paradoxical reactions described in the literature in the last ten years, which demonstrate the importance of considering paradoxical reactions in patients who present with new or worsening signs and symptoms after starting tuberculosis treatment.

## 1. Introduction

Worsening during anti-tuberculosis therapy is often synonymous with therapeutic failure. However, in some cases, the worsening of the patient’s clinical picture, as well as the appearance of new signs and/or symptoms, constitutes a so-called paradoxical reaction [1]. These reactions are defined by clinical or radiological worsening of pre-existing tuberculous lesions or the development of new lesions in patients receiving anti-tubercular drugs [2]. It is a phenomenon that can severely complicate a patient’s recovery, potentially leading to further morbidity and residual deficits. Paradoxical reactions remain poorly understood regarding their pathophysiology and management [3]. These reactions are more commonly observed in patients coinfected with human immunodeficiency virus (HIV) after the initiation of antiretroviral therapy (ART) [4]. This immune reconstitution syndrome can also occur in immunocompetent individuals, and the syndrome includes a variety of manifestations, such as intestinal perforation, vertebral tuberculosis, tubercular liver abscess, retropharyngeal abscess, endobronchial obstruction, skin abscess and various other syndromes [5]. This phenomenon is usually diagnosed by excluding other causes of TB worsening, such as drug resistance, treatment failure and adherence issues. Paradoxical reactions represent an important challenge for clinicians, as they inevitably lead to prolonged hospitalization. We report a case of a 25-year-old immunocompetent man who suffered paradoxical worsening of disseminated TB during anti-TB treatment manifested as osteomyelitis of the foot and disseminated subcutaneous abscesses. Finally, we report a review of the literature regarding cases described of paradoxical reactions in the last ten years.

## 2. Case Presentation

A 25-year-old man reported onset of fever, cough and weight loss starting in June 2024. He is an immigrant from Bangladesh and arrived in Italy in November 2023, where he lives in a community together with other immigrated people. However, before arriving in Italy, he faced a long journey, crossing several countries, including Libya, where he spent a certain period in prison with poor sanitary conditions. On 4 July 2024, he went to the emergency room, complaining of a fever and cough. Blood tests were performed and showed anemia (Hb 7.4 g/dL), which required blood transfusions. The patient’s blood count showed no alterations in the leukocyte formula: WBC, 5380/mmc; neutrophils, 3500/mmc; and lymphocytes, 1200/mmc. Renal and liver function were within normal limits (creatinine, 0.94 mg/dL; AST, 31 U/L; ALT, 30 U/L), with altered inflammation biomarkers (C-reactive protein, 7.06 mg/dL). An HIV screening test was performed, with a negative result. A CT scan of the chest and abdomen was also performed, with findings of multiple colliquated lymphadenopathies in the mediastinum, thickening with cavitation in the apical segment of the right upper lung lobe and multiple hypodense lesions of the spleen (Figure 1). The interferon–gamma release assay (IGRA) was positive. Subsequently, fibrobronchoscopy was performed, with a positive result for *M. tuberculosis* upon direct examination, polymerase chain reaction (geneXpert) and cultural examination. The same result was obtained from a fine needle aspiration of a hilar lymph node. The search for resistance to first-line anti-tuberculosis drugs yielded negative results. Considering the microbiological result, the patient was admitted to the infectious diseases department of Pesaro Hospital, where antituberculosis therapy was started with 300 mg isoniazid, 600 mg rifampicin, 1200 mg ethambutol and 1500 mg pyrazinamide daily, with consequential clinical improvement and resolution of the patient’s cough and fever. On 31 July, the patient was discharged and sent home with instructions to continue anti-tuberculosis therapy, in addition to the scheduling of blood tests and check-ups.

On 14 August, the patient came for a check-up, complaining of severe pain in the left foot and ankle and difficulty walking. The patient was hospitalized again in the infectious diseases department of Pesaro Hospital in order to carry out a series of tests.

On 4 September, the patient underwent a magnetic resonance imaging scan of the foot, with findings of osteomyelitis affecting the cuboid bone (Figure 2 and Figure 3). A fine needle aspiration of ankle-joint fluid was performed, which showed a positive result for Mycobacterium tuberculosis. In consideration of the widespread tuberculosis (lymph nodes, lungs and bones), the anti-tuberculosis therapy was strengthened with a fifth antibiotic with good penetration into the bone tissue, i.e., 750 mg of levofloxacin daily, with subsequent resolution of pain and resumption of normal walking.

Within a few days, the patient showed the appearance of subcutaneous swellings at the left periareolar, right subareolar and right paraumbilical levels (Figure 4). Subsequently, swellings also appeared at the right supraclavicular and left subscapular levels. The swellings appeared with varying degrees of size, erythema and pain. At the level of the left periareolar swelling, a needle aspiration of purulent fluid was performed, with a positive result again for *M. tuberculosis* and same good drug-sensitivity profile.

The anti-tuberculosis therapy was continued with five drugs (isoniazid, rifampicin, ethambutol, pyrazinamide and levofloxacin) for a period of 6 months, and subsequently, the therapy was continued with two drugs (isoniazid and rifampicin) and it is still in progress. In January 2025 and February 2025, the skin and subcutaneous swellings were repeatedly drained, with the absence of *M. tuberculosis* upon bacteriological examination. Considering the persistence of skin swellings, we decided to start steroid therapy with prednisone at an initial dosage of 25 mg bid for a week, followed by subsequent slow tapering with final resolution of the clinical picture.

## 3. Literature Review

Several cases of immunocompetent people with tuberculosis infection who developed a paradoxical reaction during treatment are described in the literature. Here, we resent cases of paradoxical reactions described in the literature in the last ten years (2014–2024) concerning immunocompetent adult subjects with tuberculosis infection. We used PubMed as a search engine. We used the words “paradoxical reaction”, “immunocompetent”, “adult” and “tuberculosis infection” as search terms. We subsequently excluded all clinical cases that did not meet our inclusion criteria (e.g., clinical cases concerning tuberculosis infections in immunosuppressed subjects or cases of tuberculosis infection in pediatric patients). We found eight cases of paradoxical reactions following the start of anti-tuberculosis therapy in immunocompetent adults (Appendix A) [5,6,7,8,9,10,11]. One of these cases concerns a 19-year-old woman with a significant state of malnutrition. The woman showed cervical lymphadenopathy fistulized, cough, fever and weight loss. Microbiological tests showed positivity for Mycobacterium tuberculosis, so therapy was started first with isoniazid, rifampicin, ethambutol and pyrazinamide, then isoniazid, rifampicin, ethambutol and moxifloxacin for initial alteration of liver function. After 4 weeks of therapy, the woman showed the appearance of skin abscesses and new lymphadenopathies. Microbiological tests conducted on the new lesions showed positivity for Mycobacterium tuberculosis; therefore, the anti-tuberculosis therapy was continued for 10 months, with final resolution of the clinical picture [5]. In another case, a 39-year-old man with a history of alcoholism presented with pulmonary and meningeal tuberculosis. During therapy with isoniazid, rifampicin, ethambutol, pyrazinamide and dexamethasone, the patient developed osteoarticular tuberculosis in the thoracic spine. Anti-tuberculosis therapy was continued for 12 months, with clinical resolution [7]. Another case describes a 26-year-old man with pulmonary tuberculosis who developed intestinal tuberculosis during anti-tuberculosis therapy. In this case, surgery and the continuation of medical therapy made it possible to achieve complete recovery [8].

## 4. Discussion and Conclusions

A worsening of symptoms during tuberculosis, with the appearance of new signs and/or symptoms, generally implies therapeutic failure. This occurs in the case of failure adhere to therapy, in the presence of drug-resistant *M. tuberculosis* or in immunosuppressed patients (for example, patients with HIV infection [12]). However, in some cases, this phenomenon, defined as a paradoxical reaction, can also occur in otherwise healthy patients and often does not require additional therapy. Similar pictures have been described in the literature—in particular, linked to forms of lymph node tuberculosis (LNTB). The prevalence of paradoxical reactions associated with LNTB varies from as low as 13.3% to as high as 35.3% [13]. Paradoxical reaction may occur during anti-tubercular treatment or be reported even after completion of treatment—called post-therapy paradoxical reaction. The onset of a paradoxical reaction may occur within a month of therapy or as long as 12 months after the initiation of an anti-tubercular drug. It is important to consider a paradoxical reaction when other causes of clinical or radiological worsening have been ruled out [14]. In conclusion, we report the case of a paradoxical reaction in a healthy patient without risk factors for immunosuppression. A paradoxical reaction must be taken into consideration as a differential diagnosis in all situations in which the patient shows worsening of the clinical picture despite ongoing therapy. In these situations, it is essential to continue anti-tuberculosis therapy despite apparent clinical worsening. The help of anti-inflammatory therapy such as steroid therapy may be necessary to control clinical manifestations in selected cases. However, further studies are needed to define the pathogenetic mechanisms underlying paradoxical reactions to find therapeutic targets to counteract this phenomenon.

## Figures and Tables

**Figure 1 idr-17-00046-f001:**
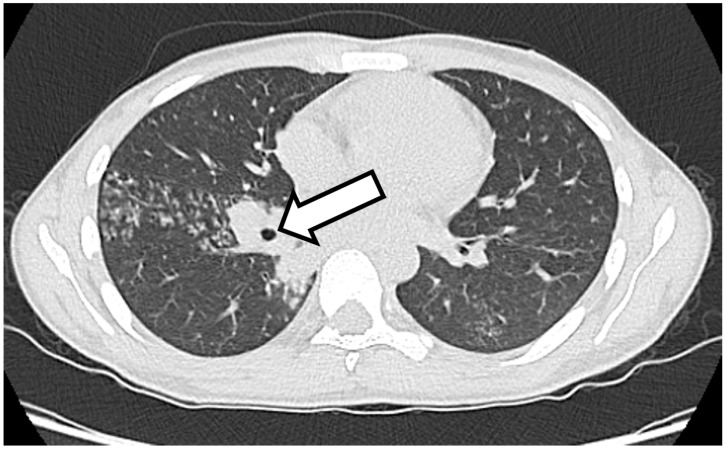
High-resolution chest CT: at the level of the apical segment of the right upper lung lobe, it is possible to see a thickening with contextual cavitation where the arrow is present.

**Figure 2 idr-17-00046-f002:**
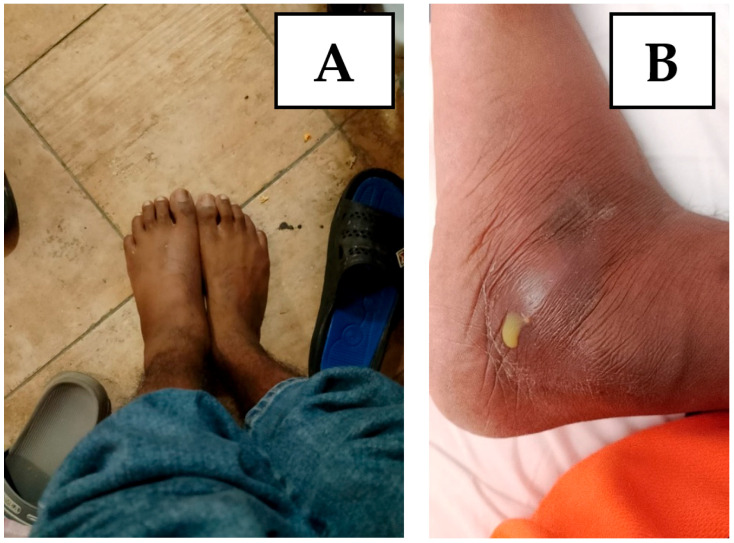
(**A**) The figure shows the appearance of swelling of the left foot compared to the contralateral one. (**B**) The figure shows leakage of caseous material from the swelling of the left foot following an exploratory puncture.

**Figure 3 idr-17-00046-f003:**
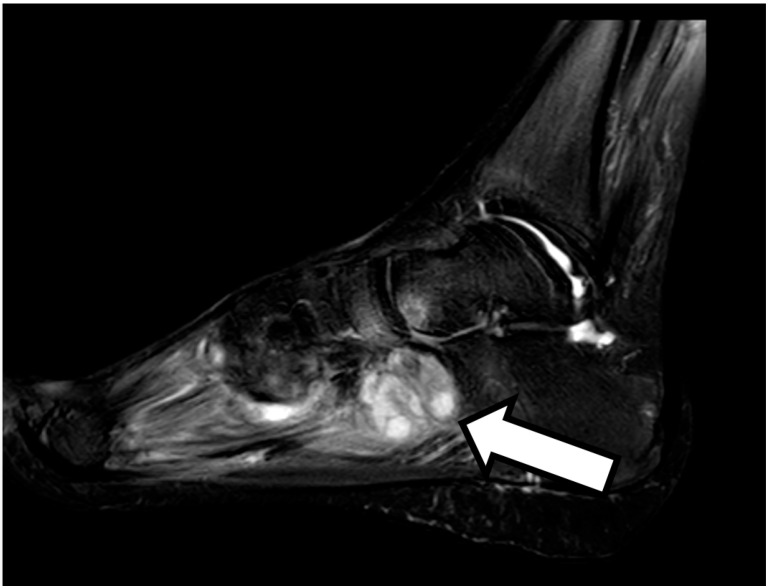
Magnetic resonance imaging with contrast medium of the left foot showing the presence of osteomyelitis affecting the cuboid bone where the arrow is present.

**Figure 4 idr-17-00046-f004:**
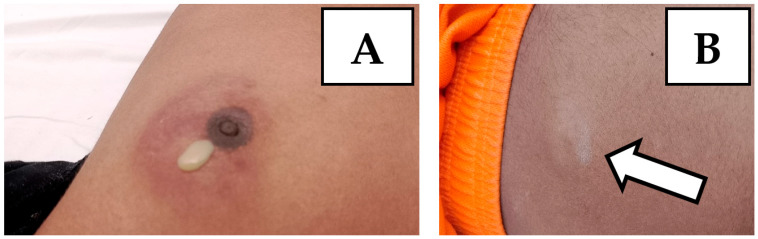
(**A**) The figure shows an abscess at the left periareolar level with spontaneous leakage of caseous material. (**B**) The figure shows a skin abscess at the abdominal level, where the arrow is present, that arose approximately 2 months after the start of anti-tuberculosis therapy.

## Data Availability

The data are available at the Infectious Diseases department of the Pesaro hospital of the Pesaro and Urbino Territorial Health Authority.

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
