# Peer review of "The Appearance of Osteomyelitis of the Foot and Disseminated Subcutaneous Abscesses During Treatment for Disseminated Tuberculosis Infection in an Immunocompetent Patient: Case Presentation of a Paradoxical Reaction and Literature Review"

_2036-7449, 2025, doi:10.3390/idr17030046_

Round 1

Reviewer 1 Report

Comments and Suggestions for Authors

I have reviewed the article and fulfilled all the requirements for a case report for publication. However, there are two points I believe should be modified:
- According to the guidelines of the International Committee on Systematics of Prokaryotes (ICSP), bacterial names should be written in italics. The full name of the microorganism is written the first time, and subsequently, the genus name can be abbreviated with the first letter capitalized and always in italics. Thus, Mycobacterium tuberculosis would be written Mycobacterium tuberculosis the first time, and subsequently M. tuberculosis.

- The acronym MRI appears on line 99 without the full name being written first.

Author Response

Comments 1: I have reviewed the article and fulfilled all the requirements for a case report for publication. However, there are two points I believe should be modified:
- According to the guidelines of the International Committee on Systematics of Prokaryotes (ICSP), bacterial names should be written in italics. The full name of the microorganism is written the first time, and subsequently, the genus name can be abbreviated with the first letter capitalized and always in italics. Thus, Mycobacterium tuberculosis would be written Mycobacterium tuberculosis the first time, and subsequently M. tuberculosis.

Response 1: I made the changes as yours instructions

Comments 2: - The acronym MRI appears on line 99 without the full name being written first.

Responde 2: I replaced the acronym MRI with magnetic resonance imaging and underlined this change in the text.

Reviewer 2 Report

Comments and Suggestions for Authors

This article address an interesting topic such as paradoxical reaction in immunocompetent patient traited for tubercuslosis.

In general the article is well written but not always easy to read and need some english improvement. 

For example:

  • From Line 36 till line 43 the term paradoxical reactin has been repeated 4 times. May I suggest to use another term (immune reconstitution syndrome..).
  • Line 59 : " he went to the emergency room where blood test ...I suggest to improve english writing:

    He went to emergency room complaining with..... Blood test were performed and showed...--A CT scan was also performed and ...

I also sugggest to use the definition of Paradoxical reaction " Paradoxical reactions are defined by clinical or radiological worsening of pre-existing tuberculous lesions, or 
the development of new lesions, in patients receiving antitubercular drugs .."  taken from https://doi.org/10.1007/s15010-024-02310-0 -Update the reference then.

Here, as follow, I will number my points and refer to line numbers in the manuscript when making specific comments

  • Line 68-76 : I think you could better explain the microbiological finidg : culture? slid medium? Migit? GenXpert result? DST?
  • Line 100 Why did you choose to prolonge ETB use ? 
  • Line 126 : there is repetition of isoniazid and rifampicin
  • Line 127 it is not clear to me for how long the patient has been treated. Up to february 2025?

In the discussion section it could be useful to address

  • the use or not of antinflammatory drugs or steroids compared to other cases in the literature.
  • If there was the need to stop and restar the treatment and if there was a need of investigate for superinfection or lack of adherence,

Finally I wonder also if there is any follow up after treatment completion because some case present paradoxical reaction also after the end of the treatment.

I think the quality of presentation can be improved also in the table making it more clear and easy to understand.

Thank you for your reply.

Author Response

Comment 1: From Line 36 till line 43 the term paradoxical reactin has been repeated 4 times. May I suggest to use another term (immune reconstitution syndrome..).

Response 1: I changed this aspect as advised

Comment 2: Line 59 : "he went to the emergency room where blood test ...I suggest to improve english writing: He went to emergency room complaining with..... Blood test were performed and showed...--A CT scan was also performed and ...

Response 2: I changed this aspect as advised

Comment 3: I also sugggest to use the definition of Paradoxical reaction " Paradoxical reactions are defined by clinical or radiological worsening of pre-existing tuberculous lesions, or the development of new lesions, in patients receiving antitubercular drugs .."  taken from https://doi.org/10.1007/s15010-024-02310-0 -Update the reference then.

Response 3: I  inserted the suggested article when I describe about paradoxical reaction

Comment 4: Line 68-76 : I think you could better explain the microbiological finidg : culture? slid medium? Migit? GenXpert result? DST?

Response 4: I decribed the way we used to discover Mycobacterium tuberculosis

Comment 5: Line 100 Why did you choose to prolonge ETB use ? 

Response 5: we prolonged the TB-therapy because of the lack of a clinical response

Comment 6: Line 126 : there is repetition of isoniazid and rifampicin

Response 6: I deleted the repetition

Comment 7: Line 127 it is not clear to me for how long the patient has been treated. Up to february 2025?

Response 7: II have looked into this aspect further. The patient is still undergoing treatment with anti-tuberculosis therapy. In addition, steroid therapy has been added to which he has responded clinically in a satisfactory manner.

Comment 8: the use or not of antinflammatory drugs or steroids compared to other cases in the literature. If there was the need to stop and restar the treatment and if there was a need of investigate for superinfection or lack of adherence. Finally I wonder also if there is any follow up after treatment completion because some case present paradoxical reaction also after the end of the treatment.

Response 8: I modified the discussion with the analyses of these aspects

Comment 9: I think the quality of presentation can be improved also in the table making it more clear and easy to understand.

Response 9: I tried to improve the presentation of the table

Reviewer 3 Report

Comments and Suggestions for Authors

The case was presented adequately.

There is no mention of HIV testing.  This must be stated.  Lymphocyte subsets are very helpful in explaining the phenomenon.  

Paradoxical reactions (probably related to immune reconstitution) is the likely mechanism behind the phenomenon.  The lymphocyte count was 1200.  What were the lymphocyte subset percentages (CD4 and CD8).  Typically one sees a low CD4 percentage that becomes higher during therapy.  For example, less than 13% CD4 that rises because of nutrition or other factors (HIV treatment).  When the percentage rises enough, for example > 20% CD4, immune responses lead to inflammation where antigen exists (M. tuberculosis organism (alive or dead).  Sometimes the sheer volume of mycobacteria exert an inhibitory effect on the T cell function and when mycobacterial load decreases to a certain level, T cell responses lead to inflammation.

Did the cultures taken from the ankle grow (presumably yes) or were the samples sterile?

Where drug levels obtained?  Were pill counts done?  Was the patient taking the medications.

What happened to the lung lesion over the time that the ankle lesions occurred?

Author Response

Comment 1: There is no mention of HIV testing.  This must be stated.  Lymphocyte subsets are very helpful in explaining the phenomenon.  

Response 1: HIV testing was negative. I added this aspect in the manuscript

Comment 2: Paradoxical reactions (probably related to immune reconstitution) is the likely mechanism behind the phenomenon.  The lymphocyte count was 1200.  What were the lymphocyte subset percentages (CD4 and CD8).  Typically one sees a low CD4 percentage that becomes higher during therapy.  For example, less than 13% CD4 that rises because of nutrition or other factors (HIV treatment).  When the percentage rises enough, for example > 20% CD4, immune responses lead to inflammation where antigen exists (M. tuberculosis organism (alive or dead).  Sometimes the sheer volume of mycobacteria exert an inhibitory effect on the T cell function and when mycobacterial load decreases to a certain level, T cell responses lead to inflammation.

Response 2: Lymphocyte typing was performed only several months after starting anti-tuberculosis therapy showing CD4 T-lymph 412/mmc (46.4%), CD8 T-lymph 229/mmc (25.8%), CD4/8 T-lymph 1.8

Comment 3: Did the cultures taken from the ankle grow (presumably yes) or were the samples sterile?

Response 3: The culture from the ankle showed positiv result

Comment 4: Where drug levels obtained?  Were pill counts done?  Was the patient taking the medications.

Response 4: Unfortunately, our availability of services did not allow us to perform a serum dosage of anti-tuberculosis drugs or an actual count of the tablets. We conducted several interviews with the patient to remind each time the importance of rigorously carrying out the therapy. The patient always confirmed that he had carried out the therapy precisely.

Comment 5: What happened to the lung lesion over the time that the ankle lesions occurred?

Response 5: the lung lesion showed resolution

Round 2

Reviewer 3 Report

Comments and Suggestions for Authors

The high percentage of CD4 T-cells noted by the authors several months after initiation of therapy is consistent with the inflammatory reaction seen.

Mycolic acid (in the cell wall of mycobacteria) has immune modulating properties.  In general, high levels of mycolic acid can have an immune suppressive effect and can contribute to inhibition of lysosomal fusion in host cells delaying killing of Mycobacteria.  As clearance of organisms occurs, the delayed inflammatory reaction (paradoxical reaction) can be manifested in areas where mycobacteria have been present all along.